# The influence of resource use on yield versus sale price trade-off in Australian vineyards

**Bryce Boyd**[1,2,3]*, **Kate Helmstedt**[1], **Mardi Longbottom**[2]*, **Kerrie Mengersen**[1]

**1** Faculty of Mathematical Sciences, Queensland University of Technology, Brisbane, Queensland, Australia, **2** Australian Wine Research Institute, Adelaide, South Australia, Australia, **3** Food Agility CRC, Sydney, New South Wales, Australia

* Bryce.polley@awri.com.au (BB); Mardi.longbottom@awri.com.au (ML)

**Data availability statement:** Data cannot be shared publicly because of legal reasons regarding sensitive finance and identity

## Abstract

Strategies for achieving sustainability in the winegrowing industry require balancing resource investment against the economic outcomes of resultant yields and sale price of the produce. Although there has been much research into forecasting outcomes and resource use, little has been done to illustrate their effects on one another and the consequential economic outcomes. This analysis uses statistical models to observe relationships between resource use, yield, and sale price. The dataset used for this analysis includes data collected for the past 10 years from 1261 vineyards located over a diverse range of Australian winegrowing regions. Yield and sale price were evaluated regarding resource use factors, such as water use and Greenhouse Gas (GHG) emissions. The analysis confirmed a strong relationship between area and resource use, with the overall area of a vineyard and its access to resources significantly determining the upper limit of yield. However, the area was also negatively related to the average sale price of grapes; we find that higher average sale prices were connected to high resource inputs per area rather than to the overall expenditure of resources. Regional and temporal effects on vineyard yield and average sales price were also identified. The analysis highlighted the importance of considering a vineyard's business goal, region, external pressures, and economies of scale when pursuing higher yields verse higher average sales prices.

## Introduction

The focus on sustainability in agronomic industries has changed how enterprises do business. The diverse nature of vineyard operations results in complicated interactions with both environmental and economic outcomes. The interactions between different factors can put an environmentally advantageous decision at odds with an economic one, and conversely, financial decisions can be detrimental to the environment [1]. As a result, many studies have included more diverse data and scales. However, studies tend to focus on specific aspects, such as reducing chemical use or improving soil health, often focusing on singular relationships or variable outcomes [2–4]. Other analyses, look at life cycles or cost-benefit analyses, reflecting a specific management decision's outcome [5–7] or their cost benefits by utilising

information. Data are available from the Australian Wine Research Institute (contact via https://sustainablewinegrowing.com.au/about-us/contact-us) for researchers who meet the criteria for access to confidential data. Public sections of the data, collected by Wine Australia are available in dashboard form at https://marketexplorer.wineaustralia.com/vintage-survey. Reports for this data as referenced in the methods are all available at https://www.wineaustralia.com/market-insights/national-vintage-report.

**Funding:** The corresponding author receives a scholarship stipend as part of their PhD program. This is funded by the Food Agility CRC. The funders had no role in study design, data collection and analysis, decision to publish, or preparation of the manuscript.

**Competing interests:** ML is employed by Sustainable Winegrowing Australia. and that This does not alter our adherence to PLOS ONE policies on sharing data and materials.

different scenarios to assess specific practices [8,9]. It has been noted that some studies have relied too heavily on single points of engagement when relaying the idea of sustainability and vineyard outcomes, especially when the inclusion of more variables or data could have better described a situation as a whole [10,11]. The changing nature of sustainability also calls for measures to be more than simple snapshots of data [10–12]. Given recent studies there is still growing interest from the winegrowing industry in further studies, with several recent reviews calling for more research to demonstrate the potential trade-offs between different operational approaches across various climates and regions [10,12,13].

A dilemma exists across agriculture through shared fundamental considerations of resource use such as water and fuel, and the resultant yield and crop value that are produced [14–16]. The average price of grapes for wine production is driven by its integration within the wine industry. An essential connection between grapes and their price is the grape's perceived quality, which initially defines a wine's potential through the grape's chemical makeup [17,18]. Grapes of higher perceived quality or grapes from well-known regions are likely to have higher prices [19]. Grape quality is connected to the market value of wine-grapes, with Wine Australia explicitly defining grape quality through the use of discrete price brackets in their annual reports [20]. It is important to note that the generalisation made to reflect quality through average price assumes the due diligence of those purchasing the grapes [21]. The economic sustainability of a vineyard is tied to this market culture, driven by the wine industry. The consideration of sustainability within viticulture is also subject to environmental and sociodemographic pressures [11]. In the Australian context, these pressures include biosecurity, climate and international market changes [22–24].

There is an extensive amount of research into the varied effects of factors on grape quality and yield [3,25,26]. Due to the lack of long-term and in-depth data there is a lack of research on grape sale price and its driving factors. Furthermore, individual factors are often studied in isolation, with yield and sales price not appearing together [27]. The lack of consolidated datasets restricts gaining statistical insights at large scales and across multiple regions. As a result, broader studies are lacking [28,29]. The dataset used for this analysis includes data spanning 10 years from a multitude of vineyards located over a diverse range of Australian winegrowing regions. We use this dataset to describe the relationship between the resources used, such as water and fuel use, with the output yield and average sale price of the resultant product, taking into account the size and location of the vineyard. In a practical sense, this aim is a baseline for comparison: given a vineyard within Australia, one could estimate the comparative efficiency of the trade-off between invested resources, yield and average sale price. This is the first time such a trade-off has been explicitly confirmed across varying regions, scales and climates in the Australian winegrowing industry.

We aim to contribute to understanding resource-use efficiency in the Australian wine industry by providing and analysing these baselines using this unique set of regional and temporal data on vineyard resource allocation, grape yield, and sales price. Our analyses reveal that resource input per area, rather than overall expenditure, is a more accurate predictor of high sales prices, especially in regions with specific climate and size characteristics. We further aim to link this effect to regional attributes, where climatic factors interact with operational scale. These contributions will help to establish a baseline for vineyard managers and policymakers to assess resource trade-offs, ultimately supporting more sustainable and economically viable decision-making in the face of climate and market pressures.

## Methods

### Data

Data used in these analyses were obtained from Sustainable Winegrowing Australia and Wine Australia. Sustainable Winegrowing Australia is Australia's national wine industry sustainability program, which aims to facilitate grape growers and winemakers in demonstrating and improving their sustainability [30]. Wine Australia is an Australian Government statutory authority governed by the Wine Australia Act [31]. Data collected by Wine Australia are publicly available. This analysis's response (predicted) variables were yield, defined as the total tonnes of grapes harvested and the average sale price of grapes in Australian dollars per tonne. Both response variables were examined as totals and also relative to the area harvested (see Table 1). Values were compared in this manner to observe how economies of scale affect the use of resources.

Data recorded by Sustainable Winegrowing Australia were entered manually by winegrowers voluntarily using a web-based interface. Data was recorded after the Australian crush (February to April), with entries due for the season by the $31^{st}$ of August each year. The SWA user manual outlines details surrounding the web interface and its requirements [32]. Vineyards volunteered data as part of Sustainable Winegrowing Australia initiatives, which included collaboration, workshops and certification of sustainable practices using third-party auditors. Vineyards were only included for each model if they recorded all the variables used for the corresponding model (see Table 1). Each vineyard had at least recorded region, harvest year, yield and area harvested. Other variables used but not present for every vineyard were average sale price, water used and fuel used (diesel, petrol, biodiesel and LPG). Fuel use was converted to equivalent tonnes of carbon dioxide and collectively referenced as emissions to enable direct comparisons between fuels. All variables were continuous except for harvest year and region, which were categorical variables (Table 1).

As data from Sustainable Wine Australia were voluntarily given, missing values were improved using regional average prices from the Wine Australia (previously the Winemakers Federation of Australia) data. Data from Wine Australia were collected via phone surveys and included total tonnes purchased, the average cost per tonne, and yearly change in price for region and grape varietal; the data is publicly available through the Wine Australia Annual reports [19,20,33–41].

**Table 1. Summary of models; their predictors, covariates and variable interactions.**

|  | Response | Predictors | Covariates | Interactions |
|---|---|---|---|---|
| **Model 1** | Yield | Water Used scope one Emissions | Area Harvested Year GI Region | N/A |
| **Model 2** | $\frac{Yield}{Area\ Harvested}$ | Water Used scope one Emissions | Area Harvested Year GI Region | Area Harvested * scope one Emissions Area Harvested * Water Use Year * Region |
| **Model 3** | Yield×Average Sale Price | Water Used Scope One Emissions | Area Harvested Year GI Region | N/A |
| **Model 4** | Average Sale Price | Water Used Scope One Emissions | Area Harvested Year GI Region | Area Harvested * Scope One Emissions Area Harvested * Water Use Year * Region |
| **Model 5** | Average Sale Price | Water Used Scope One Emissions | Year GI Region | Year * Region |

The dataset was split into two subsets that could be used across the different models. The first subset contained all vineyards and was used for two models (Model 1 and Model 2, see Table 1). The second subset contained vineyards that either recorded a value for an average price of sale per tonne through Sustainable Winegrowing Australia or were within a region with an average price of sale recorded by Wine Australia; this subset was used for three further models (Models 3, 4 and 5, see Table 1). These subsets meant that the data would be limited to samples with recorded values for the response variables (see Table 1), as every sample/vineyard had a recorded value for yield but not the average sale price per tonne.

The first subset of data (used for Model 1 and Model 2, see Table 1) contained 5298 samples spanning 2012 to 2022, covering 55 GI Regions and 1261 discrete vineyards.

The second subset of data (used for Model 3, Model 4 and Model 5, see Table 1) contained 2878 samples spanning the period from 2015 to 2022, covering 51 GI Regions and 944 separate vineyards. The average sale price per tonne was extracted from Wine Australia (1842 values) and Sustainable Winegrowing Australia (remaining 1036 values).

Additional variables were considered for analysis but were excluded due to being either underreported or had insignificant contributions to model accuracies. Variables explored but not used due to low reporting values included fertiliser (kg), electricity (Kw/h), and scope two emissions (TCO$^2$ equivalent). Variables were considered but ultimately removed due to a lack of significant contributions to models, including using renewable energy, contractors, and pressures such as frost, fire, and disease.

Data preprocessing was conducted before analysis using the Python programming language [42]. Preprocessing included the conversion of fuel to scope one emission and prior calculations for all continuous variables, including logarithmic transformations, centring and scaling by standard deviation. We converted multiple emission sources into scope one emissions using the equation from the Australian National Greenhouse Accounts Factors [43]. The calculation was conducted using

$$tCO_2e = \frac{Q \times EC \times EF1 + EF3}{1000},\qquad(1)$$

where emissions were the product of the quantity of fuel in litres, $Q$, a prescribed Energy Content, $EC$, and emission factors (as given by the Australian National Greenhouse Accounts factors) of scope one, $EF1$, and scope three, $EF3$, to tonnes of Carbon Dioxide Emission equivalent, $tCO2e$ [44].

Differences in vineyard locations were captured through the use of Geographical Indicator Regions defined by Wine Australia [24,45,46]. Although vineyards generally differed in income, there were pronounced differences between regions as shown in Fig 1. These differences were likely due to the site of a vineyard being important as it predetermines several physical parameters such as climate, geology and soil, making location a widely considered key determinant of grape yield and average sale price [27,47,48]. Each GI Region has its unique mixture of climatic and geophysical properties that describes a unique winegrowing region within Australia and is a protected trademark under the Wine Australia act [31]. Wine Australia and Sustainable Winegrowing Australia used the same GI Region categorical variable format to describe location.

Each GI Region's climatic properties were summarised using predefined classifications per the [32] user manual. The user manual describes climates by rainfall and temperature, creating supersets of regions of similar climatic properties. The climatic groups illustrated similarities and differences in sets larger than GI Regions. These classifications were literal descriptors used by industry members to discern between different weather types within winegrowing

Average Vineyard Income by Region

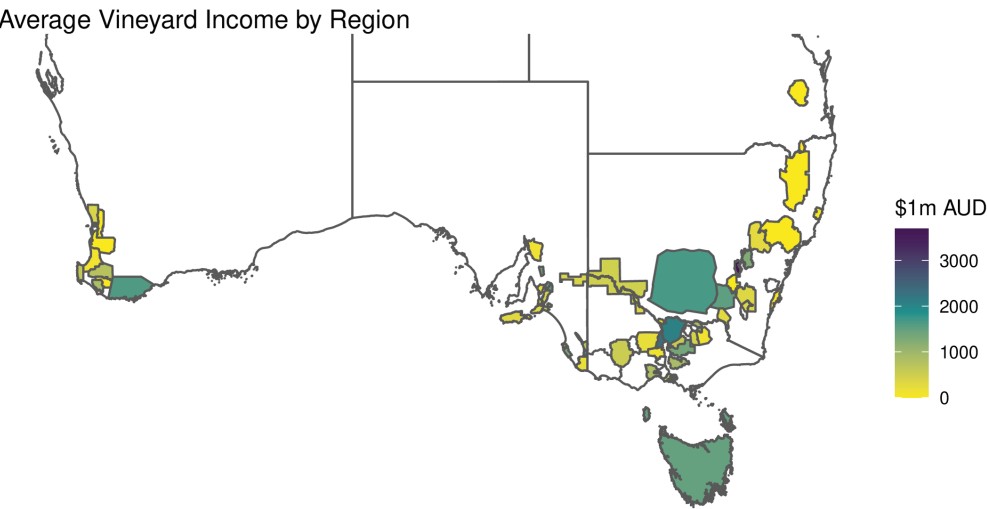

**Fig 1. Map of each GI Regions' (outlined by Wine Australia) average income for a vineyard of that region (average grape sale price per tonne × total tonnes yielded).**

regions. However, the climate descriptors for Australian GI regions were originally conceived by [49] using temperature-based indices. A key metric employed was the Mean January Temperature (MJT), which indicates heat accumulation during the growing season. Regions were categorised based on their MJT values into cool, intermediate, warm, and hot climates. This classification was used to aid viticulturists in selecting grape varieties best suited to the thermal conditions of each region. Further in-depth summarisations of Australian wine region climates can be found in greater detail using Wine Australia's climate atlas [50]. Other climatic descriptors were explored, such as the Köppen climate classification; however, this index did not offer enough granularity to highlight any trends or clustering within the data.

## Analysis

General Linear Models (GLMs) were chosen as the primary analytical approach due to their ability to quantify and interpret relationships between continuous predictor variables (such as water use and emissions) and the continuous response variables of interest (yield, yield per hectare, and average sale price). GLMs offer the flexibility to examine both main effects and interaction terms [51], which is essential for exploring the nuanced interplay between resource use, vineyard size, and regional factors. Many prior studies have used GLMs to model vineyard relationships for response variables such as yield [52,53]. Additionally, GLMs allow us to incorporate categorical variables (Year and GI Region) to account for temporal and spatial heterogeneity, making it possible to observe differences in vineyard performance across diverse Australian winegrowing regions over time. The interpretability of GLM coefficients makes this approach attractive for practical applications in vineyard management [54], providing insights into the magnitude and direction of each variable's influence on yield and sale price outcomes (given the other variables in the model).

Pairwise Pearson Correlation Coefficients (PPCC) were calculated to assess the potential existence of linear relationships between the input and predicted variables. Both the magnitude of the value and the associated confidence intervals were evaluated to determine if a coefficient was indicative of a strong relationship. P-values reflected the significance of a given

correlation coefficient, with statistical significance being declared when the corresponding P-value was lower than 0.05. PPCC analyses were undertaken for data on the original scale and data as a logarithmic transform. Transforming data before calculating the coefficients changes several things. The logarithmic transform of the data alters the interpretation of the coefficients to percentage change; a coefficient will indicate the change in percentage of one variable compared to the other; scaling by standard deviation also changes this interpretation to be a percentage of that variable's standard deviation. When considering the logarithmically transformed variables, a coefficient of 1 would indicate that the change of one variable by one percentage of its standard deviation would correlate to the other variable changing by one percentage of its standard deviation. The importance of this is the dimensionless nature of these relationships and that it can be translated directly to any vineyard's case with a well-known distribution.

Five general linear models (GLMs) were created to explore the interactions between predictors (see Table 1) and allow for easily comparable differences in the influence and magnitude of relationships. Model fit was measured using $R^2$ and adjusted $R^2$ and F statistics. T-tests were used to determine if predictors significantly contributed to the models when accounting for other variables, showing which specific years and areas contributed significantly. Both the PPCC and GLM analyses were created using the R statistical programming language [55] with the Caret package [56].

Various alternate methods were explored, including splines, hierarchical regression, general additive, and generalised linear models. These alternative approaches were not used as final models because they offered little further insights or improvements in accuracy.

## Model validation

Models were validated using K-fold cross-validation. This was performed by removing a subset of data from the sample used to train models and then predicting those variables to determine how sensitive the model is to changes in the sample data. For this analysis, each model was validated using 10 folds, repeated 100 times.

## Results

### Summary statistics

Table 2 shows the summary statistics of each variable in its original units. The range of these values shows the level of difference between some vineyards, with operations differing by orders of magnitude in size, yield and average price of sale (See Table 1). PPCC values for the transformed, centred and scaled variables are shown in Table 3. The process of centring and scaling variables changes them so that their mean is centred on 0, and all values are scaled to the variable's standard deviation, such that 1 is equal to the variable's standard deviation. All correlations were found to be statistically significant (p < 2.200E-16), except for Average Sale Price, all variables were positively correlated. Water use, area harvested, and emissions being positively correlated to yield, it can be considered that more resources and area are likely to lead to greater yields. The negative correlations between Average Sale Price and Yield, Water Use, Area and Scope One Emissions indicated that Area Harvested and Fuel Use were not the determining factors for average sale price. The negative correlations are associative, not causal relationships (i.e. using more water does not cause lower sale prices).

**Table 2. Summary statistics of each continuous variable.**

| Variable | Mean | Standard Deviation | Minimum | Maximum |
|---|---|---|---|---|
| Yield (tonnes) | 7.76E+02 | 2.18E+03 | 1.00E+00 | 7.23E+04 |
| Area Harvested (ha) | 6.67E+01 | 1.34E+02 | 7.00E-02 | 2.44E+03 |
| Water Used (ML) | 7.47E+06 | 5.65E+08 | 1.00E+00 | 4.27E+10 |
| Scope One Emissions ($tCO_2e$) | 4.17E+04 | 8.57E+04 | 6.76E+00 | 2.11E+06 |
| $\frac{\text{Yield (tonnes)}}{\text{Area harvested (ha)}}$ | 1.01E+01 | 8.13E+00 | 4.00E-02 | 8.63E+01 |
| Average Sale Price (AUD/tonne) | 1.48E+03 | 9.22E+02 | 1.60E+02 | 2.60E+04 |
| $\frac{\text{Average Sale Price (AUD/tonne)}}{\text{Area Harvested (ha)}}$ | 1.35E+02 | 5.71E+02 | 1.75E-01 | 2.98E+04 |

**Table 3. Pairwise Pearson correlation coefficients for logarithmically transformed values.**

| | Yield | Area Harvested | Water Used | Scope One Emissions | Yield by Area | Average Price | Average Price by Area |
|---|---|---|---|---|---|---|---|
| Yield | 1.00 | 0.88 | 0.82 | 0.76 | 0.96 | -0.46 | -0.88 |
| Area Harvested | 0.88 | 1.00 | 0.78 | 0.83 | 0.73 | -0.19 | -0.81 |
| Water Used | 0.82 | 0.78 | 1.00 | 0.67 | 0.76 | -0.49 | -0.82 |
| Scope One Emissions | 0.76 | 0.83 | 0.67 | 1.00 | 0.65 | -0.16 | -0.67 |
| Yield by Area | 0.96 | 0.73 | 0.76 | 0.65 | 1.00 | -0.54 | -0.84 |
| Average Price | -0.46 | -0.19 | -0.49 | -0.16 | -0.54 | 1.00 | 0.72 |
| Average Price by Area | -0.88 | -0.81 | -0.82 | -0.67 | -0.84 | 0.72 | 1.00 |

## General linear models

Each model had a high $R^2$ value, indicating that most of the variance within the data was described by the models (see Table 4) and statistical significance of F-tests (p < 2.200E-16). Aside from 3 variables, all regression parameters were statistically significant from zero; F-tests across each model's variables were (p < 0.05). The three exceptions were scope one emissions in Model 3 (p=0.22) and Model 4 (p=0.39) and the interaction between the area harvested and water used in Model 2 (p=0.22) (with Table S1 listing the p-values of water used for each response variable). Note that scope one emissions were included in all models to directly compare the response variables as ratios of vineyard size to raw values and because it was strongly correlated to the response variable in every model (except model 5), especially for Models 1 and 4 (Table 3). The appendix provides further supporting diagnostics of the models in the form of QQ and residuals plots (Figs S1 to S10).

Table 5 summarises coefficients related to continuous variables. In Model 1, all coefficients except for the intercept significantly contributed to the model (p < 0.05), and in Model 2, all

**Table 4. Summary of models; their performance, F-statistics and Residual error.**

| | $R^2$ | Adjusted $R^2$ | F-Statistic | P-Value | Residual Standard Error | Residual Sum of Squares | Residual Mean of Squares |
|---|---|---|---|---|---|---|---|
| **Model 1** | 0.9072 | 0.9061 | 775.3 | 2.200e-16 | 0.3065 | 491.3 | 0.1 |
| **Model 2** | 0.8291 | 0.8141 | 55.07 | 2.200e-16 | 0.4312 | 905.03 | 0.19 |
| **Model 3** | 0.9753 | 0.9748 | 1885 | 2.200e-16 | 0.1589 | 71.11 | 0.03 |
| **Model 4** | 0.9091 | 0.9006 | 106.1 | 2.200e-16 | 0.3153 | 261.41 | 0.10 |
| **Model 5** | 0.9089 | 0.9004 | 107.2 | 2.200e-16 | 0.3155 | 262.04 | 0.10 |

**Table 5. Summary of each Model's coefficients for continuous variables.**

| | | Intercept | Area Harvested | Water Used | Scope One Emissions | Area Harvested ∗∗ Scope One Emissions | Area Harvested ∗∗ Water Used |
|---|---|---|---|---|---|---|---|
| Model 1 | Coefficient | -3.32E-02 | 7.42E-01 | 8.66E-02 | 6.73E-02 | | |
| | Std Error | 1.96E-02 | 1.00E-02 | 8.90E-03 | 8.00E-03 | | |
| | $Pr(>|t|)$ | 9.09e-02 | <2e−16 | <2e−16 | <2e−16 | | |
| Model 2 | Coefficient | 1.70E-01 | 5.77E-01 | 1.08E-01 | 8.50E-02 | -4.97E-02 | -5.35E-02 |
| | Std Error | 5.91E-02 | 1.48E-02 | 1.31E-02 | 1.17E-02 | 8.10E-03 | 8.40E-03 |
| | $Pr(>|t|)$ | 4.15e-03 | <2e−16 | 2.09e-16 | 5.41e-13 | 9.10e-10 | 2.66e-10 |
| Model 3 | Coefficient | 1.81E-02 | 9.71E-01 | -2.31E-02 | -7.00E-03 | | |
| | Std Error | 1.30E-02 | 7.20E-03 | 6.90E-03 | 5.70E-03 | | |
| | $Pr(>|t|)$ | 1.66e-01 | <2e−16 | 8.55e-04 | 2.22e-01 | | |
| Model 4 | Coefficient | 1.45E-01 | 2.40E-03 | -4.66E-02 | -1.70E-02 | 1.15E-02 | 1.40E-03 |
| | Std Error | 5.28E-02 | 1.50E-02 | 1.43E-02 | 1.18E-02 | 7.90E-03 | 8.30E-03 |
| | $Pr(>|t|)$ | 6.10e-03 | 8.76e-01 | 1.16e-03 | 1.49e-01 | 1.47e-01 | 8.63e-01 |
| Model 5 | Coefficient | 1.52E-01 | | -4.04E-02 | -1.71E-02 | | |
| | Std Error | 5.27E-02 | | 1.13E-02 | 9.70E-03 | | |
| | $Pr(>|t|)$ | 4.01e-03 | | 3.53e-04 | 7.97e-02 | | |

coefficients were statistically significant (p < 0.05). However, scope one emissions did not significantly contribute to models 3, 4 and 5. Model 4 only had statistically significant contributions from the intercept and water use. Although the coefficient for water use was statistically significant for each model, it did not have the highest value; instead, the area harvested, being an order of magnitude greater, dominated the models. Model 5 achieved a similar $R^2$ to Model 4 without area harvested, having stronger influences from water use and scope one emission.

The regression coefficients for the Year variable of each model are depicted in Fig 2. For each model, the first year of data was used for the model's baseline: 2012 for Models 1 and 2 and 2015 for Models 3, 4 and 5. The Adelaide Hills is used as the regional baseline for each of the models. And, for the interaction between year and region, the first year and the Adelaide Hills is used as the baseline. Region and year contributed more than the other variables in some but not all cases. However, the coefficients varied substantially over the years and were not significantly different from zero in some years. Models 4 and 5 are very similar in Fig 2, indicating that the exclusion of area does not significantly affect the contribution from annual influence. Models 4 and 5 have the most prominent trends, showing an increase in yearly effects over time, with Model 3 also increasing from 2016 to 2018 but plateauing afterwards. Models 1 and 2 do not show a clear trend but drop during 2017 and 2018 after rising in the first 3 years.

Fig 3 summarises regional differences by temperature and rainfall. The most notable difference is between vineyards within 'Hot' and 'Very Dry' regions (warm inland regions), where high average sale prices are historically low, and yield is high. Water Use changes dramatically between these regions as well, with water being a driving force for yield but not necessarily average sale price. The warmer and drier regions also tend to have larger vineyards. These regional differences are further shown in Fig 4, as a ratio of vineyard area, where there is a pronounced difference in 'Hot' and 'Very Dry' regions producing more per area with lower average sale prices per area than other regions.

Fig 4 further shows the emphasis that 'Hot' regions have on high yields, with low average sale prices compared with other regions. Scaling average price and yield by area shows a strong negative trend, trading quantity for higher sales prices.

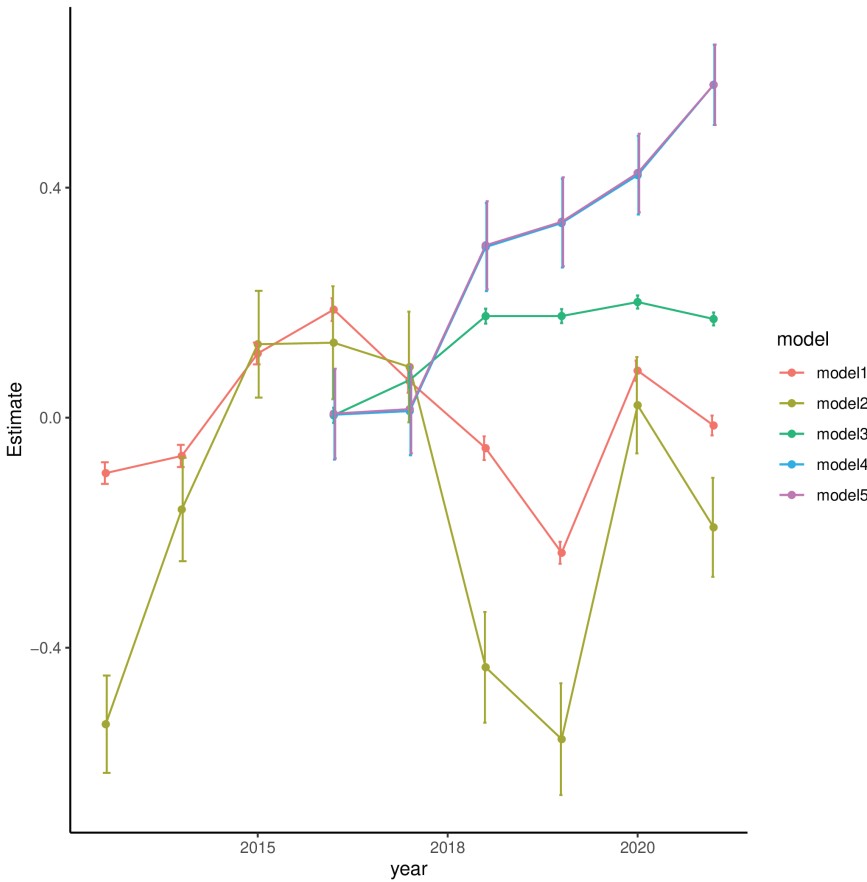

**Fig 2. Model Coefficient values for each year with standard error bars.** Each coefficient represents the influence that each year had on the model outcomes (see Table 1 for model outcomes and their predictors). Each year's influence is compared to the first year of the model.

Table 6 shows the validation results of each of the models. The $R^2$ measures of fit show similar results to the initial models, with a slight decrease as expected, indicating that the models are robust and consistent.

Fig 5 highlights two of the most prominent distinct clusters of vineyards based on climate characteristics, specifically comparing warm/hot inland regions and cooler coastal regions. The clustering reveals that warmer inland areas tend to have higher yields per hectare but at a lower profit per area, whereas cooler coastal regions show lower yields but higher profit margins per hectare. This contrast emphasizes the interplay between climate conditions and economic outcomes.

## Discussion

There was an expected strong relationship between size and resource use, with the overall area of a vineyard and its access to resources significantly determining the upper limit of potential yield. However, size was also inversely related to the potential average sale price. Higher average sales prices were also associated with high resource inputs per area instead of the overall expenditure of resources. Vineyard yields and sales prices changed considerably by region and year. Even given regional and yearly changes, there was a strong connection between smaller

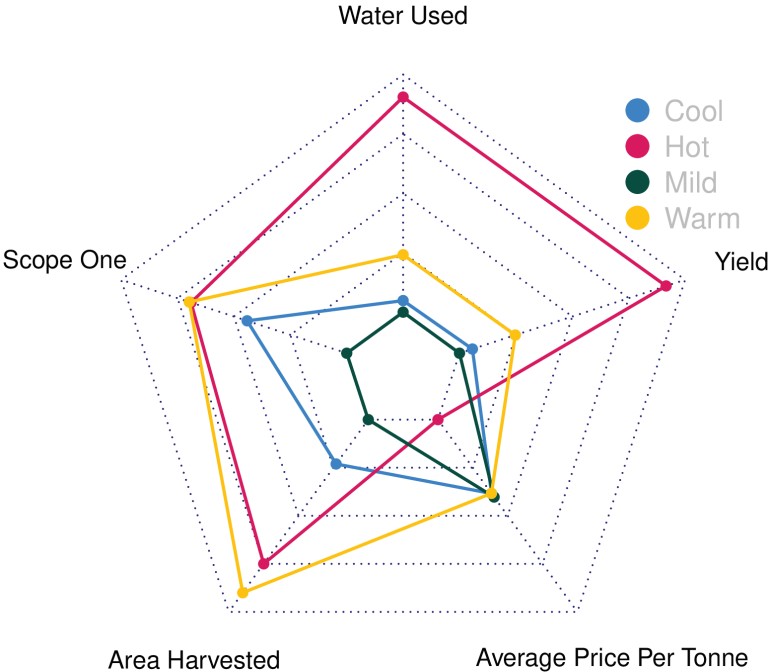

**Fig 3. Radar plot of climatic profile's resource use, yield and average sale price.** Each point shows the overall contribution of a variable to the vineyard's outcomes for that climate. The left reflects vineyards in different climatic temperatures. The right reflects vineyards in different rainfall climates.

vineyards and higher sales prices. This could have been due to more attention available when managing smaller properties. Additionally, the strong negative correlation between vineyard size and average sale price could be due to regionally differing vineyard sizes, with warmer inland regions tending to have much larger vineyard sizes.

The difference between regions can also be due to varied reasons, such as climate; where warmer regions result in a shortening of the ripening period and the risk that if harvest occurs in a duration of high temperature, there could be a negative impact on wine quality [57–59] and yield [60,61]. Climate change might move the north and south latitude boundaries of areas suitable for good quality wines [62]. It could even lead to improvements in fruit production and quality improvements in some areas [63]. However, other areas may be negatively affected by high temperatures and water stress due to reduced water availability. There is significant room to investigate the impact of different climate scenarios on these other regions and further the various consequences of climate change, specifically for water in the Australian context.

The findings from this study reveal notable differences in yield and profitability between vineyard clusters based on climate conditions, specifically between the warm/hot regions and cooler/damper regions, with warmer drier regions tending to be further inland in Australia, compared to the more coastal cooler and damper regions. Warmer areas are characterized by longer growing seasons and higher heat accumulation, and are generally associated with higher yields per hectare (as shown by the results in Figs 4 and 5). However, these conditions often entail trade-offs in grape quality, as accelerated ripening due to higher temperatures can reduce the complexity of flavours—a key determinant of premium wine quality [64,65]. Consequently, grapes from warmer regions tend to be lower-priced, with an emphasis on yield

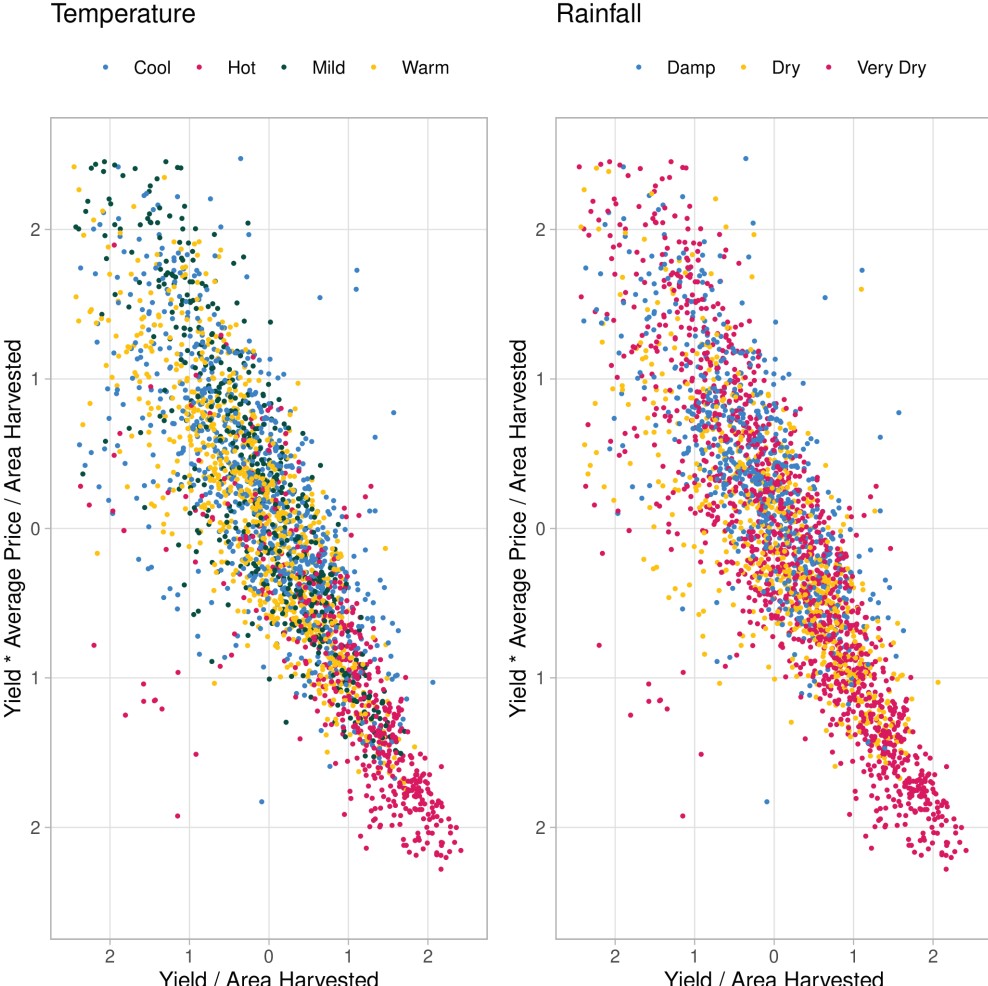

**Fig 4. Scatter plot of vineyard yield against the average sale price as ratios to area harvested.** The axes are in standard deviations as absolute values with points coloured by climate.

rather than quality [66]. In contrast, cooler coastal regions, which benefit from moderate temperatures and oceanic cooling, produce lower yields per hectare but command higher prices due to the slower ripening process that fosters more nuanced flavours in the grapes [60,65].

These climate-based clusters suggest distinct strategies for vineyard management that could enhance both economic outcomes and sustainability. Vineyards in warm/hot inland regions may benefit from water-efficient practices, such as mulching [67] to maximize yield while conserving resources, thereby sustaining profitability through volume despite lower price points per tonne. Conversely, vineyards in cool/damp coastal regions could focus on quality-enhancing practices, such as canopy managements [68], to optimize profit potential per hectare, reflecting the market demand for premium-quality grapes. Future research could build upon these findings by refining cluster characteristics with additional factors, such as vineyard size and soil type, to offer more targeted recommendations. Such an approach may further clarify the influence of specific environmental and operational factors on vineyard productivity and profitability, providing valuable insights for region-specific, sustainable vineyard management.

**Table 6. Model validation using k-fold cross validation, for ten folds repeated 100 times.**

|  | Residual Mean Squared Error | R2 | Mean Average Error |
|---|---|---|---|
| Model 1 | .309 | .905 | .2165 |
| Model 2 | .457 | .7921 | .313 |
| Model 3 | .165 | .972 | .101 |
| Model 4 | .348 | .878 | .182 |
| Model 5 | .348 | .878 | .183 |

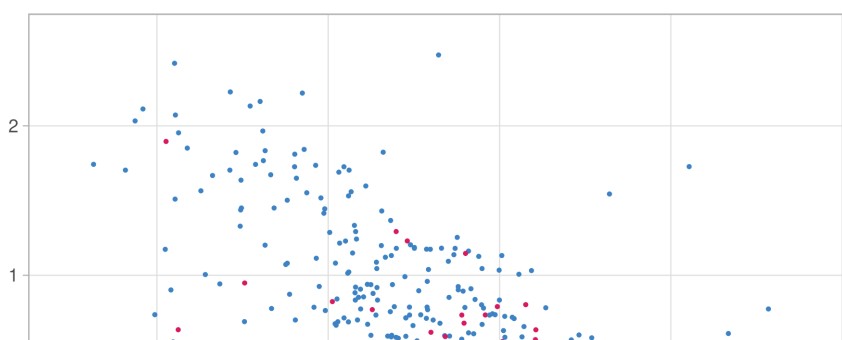

**Fig 5. Scatter plot of vineyard yield against the average sale price as ratios to area harvested.** This figure includes only vineyards from cool/damp and Hot/Dry climates, plotted against each other to aid in highlighting the division of these clusters. The axes are in standard deviations as absolute values with points coloured by climate.

Given its F-statistics, the lack of significance of scope one emissions and its contribution to models could indicate that other vineyard activities requiring fuel are not leading factors for a vineyard's average sale price. The relationship between yield, value, and area was not simply about efficiently producing the most grapes. It is possible that the relationship of scope

one emission between yield and sale price was closely tied to a vineyard's area due to requiring more fuel to address more issues over greater distances. It is difficult to directly discern the connections between scope one emissions, as fuel can be used for a broad category of activities. As [69] highlights, the interpretation of emissions is uncertain and inexplicit when comparing viticulture practices. Many factors, especially size, can directly or indirectly influence the dynamics of carbon emissions in a vineyard's life cycle. Notably, scope one emissions account for only a small portion of the total energy input in vineyards, as other significant inputs, such as fertilizers, pesticides, and electricity, collectively contribute to a substantial share of the carbon footprint in grape production [70].

There are important considerations unique to winegrowing compared to other agricultural industries. The vertical integration of winegrowing within the wine industry ties winegrowers to secondary and tertiary industries, such as wine production, packaging, transport and sales. This results in unique issues and considerations for each vineyard, where on-the-ground decisions are influenced by other wine industry choices, such as the use of sustainable practices in vineyards as a requirement for sale in overseas markets; notably, these interactions can be further complicated by some winegrowers being wholly integrated into a wine company, while others are not [29]. Incorporating decisions into the model could help describe the factors contributing to regional differences beyond resource consumption, motivating the call for more granular data and more sophisticated modelling.

Many on-the-ground decisions influence both sales price and yield. The decision to prioritise average sale price over quantity, is governed by complex physical and social forces, for example international market demands, disease pressures and natural disasters [24,71–76], with many of these occurrences being highlighted throughout the reports from Wine Australia over the past decade [19,20,33,35,37–41]. However, the changes in the coefficients (see Fig 2) are not reflective of many known occurrences, such as the 2020 bush fires, which had higher values for coefficients than prior years; During the 2020 bush fires 40,000 tonnes of grapes were lost across 18 different wine regions due to bush fires and smoke taint. Compared to countrywide pressures such as drought, this damage made up only 3% of the total amount of grapes for that year; although acknowledged as a considerable loss on an individual basis, it was deemed to be only a minor national concern by Wine Australia when compared to other environmental pressures such as drought [34]

Climatic pressures are an important consideration for growers, especially those in warmer and drier regions. The Wine Australia reports also show that warm inland regions have seen a decline in profit over the past decade, whereas regions with lower average sales prices did not [19,20,33,34,37–41]. Vineyards in warm inland regions also tend to be larger, making up for lower sale prices with larger yields. Considering the negative correlation of average price to area, economies of scale become essential for this strategy. Given the large quantities of grapes that can be produced by some vineyards, even at low margins, there is the potential to be profitable. However, the increasing climatic pressures, mixed with the requirement for larger volumes of water, make the sustainability of some vineyards come into question. Furthermore, intensive farming, in general, is known to jeopardise the sustainability of an operation through the degradation of soil and waterways [77–79]. There are established methods that can help to mitigate these effects, such as the use of cover crops, midrow crop rotation and efficient irrigation.

Some regions appeared to produce grapes at lower average sale prices at scale, while others produced higher in lower volumes. This empirical finding is consistent with Wine Australia's annual reports, which shows that some GI regions are known for producing large amounts of lower grade (low value per tonne) grapes [35,41]. Comparatively, other regions

only produce grapes of higher sales price but in smaller quantities. The difference in pricing per tonne between the lowest and highest regional average sales prices was almost a hundred times, showing that region had a profound influence. Some regions also had a mixture of high and low average sales prices and yield, showing regional variability in pricing, which may be explained by the varieties produced. A further possibility is the existence of regional upper limits on the potential sales price or that there are diminishing returns in some regions when pursuing higher sales prices or quantity; however, these types of relationships may be obfuscated by knowledgeable winegrowers who avoid such pitfalls.

Due to regional differences, different strategies are employed, such as some regions targeting mass production over higher sales prices. This is most notable when grouping regions by climate, especially when considering GI Regions in the 'Hot Very Dry' climate (see Fig 4). Fig 3 also shows that comparatively, 'Warm' and 'Dry' Regions manage their resources incredibly efficiently, having generally larger areas using similar resources to those in 'Cool' regions but having areas comparable to regions in 'Hot' climates. The coefficients for Model 4 also show a more significant benefit of resource use per area when producing grapes at higher average sale prices, showing higher resource use per area reflective of the higher average sales price (see Table 2). Although not chosen over the GI region, climate was considered a significant determinant of the ability to produce larger quantities of grapes and a determinant in grape sale price [47]. The more granular GI Region likely explained a broader mix of geographical phenomena, such as soil, geology and access to water resources [27,80]. The interaction between the YYearand the GI Region likely accounted for events such as bushfires, which would be impactful, but only locally, both in time and space.

For winegrowers, understanding the inverse relationship between vineyard size and average sale price suggests that smaller vineyards may benefit from focusing on high-value, lower-yield practices, which allow for a more targeted approach to quality management. In larger vineyards, where economies of scale may drive profitability, strategies that optimise resource use efficiency per hectare become crucial, especially in regions facing water scarcity. Here, larger vineyards could benefit from adopting further sustainable and climate adaptive practices such as mulching [67], vineyard canopy and mid-row management strategies [68], and soil nutrient leaching and organic management strategies [71,81,82] to help aid in water, fuel and fertiliser reductions. Given the climatic pressures in warmer inland regions, especially those with high water requirements, these practices may also mitigate the risk of crop failure due to drought or heat stress, enhancing both resilience and economic viability. Additionally, international market trends increasingly favour sustainably sourced wine, suggesting that supporting vineyards in sustainable water and fuel management could enhance export opportunities, meeting both environmental goals and market expectations.

We identified two main limitations to our linear modelling. First, models 1 and 2 overpredicting yield may have been due to preventative measures brought on by regional pressures such as fire, frost and disease. More fuel and water were likely used to prevent these issues from spreading within a region, thus disproportionately affecting some vineyards compared to others locally. This type of maintenance is not well captured in the models, especially when some regions, especially those in warmer regions, are not as prone to disease as cooler climates and could potentially have lower fuel and water use per hectare. This could create a discrepancy in vineyards that utilised preventative measures in wetter regions, as opposed to those that did not, thus expending less fuel and energy but risking disease. When reviewing the differences between regions, it is essential to consider that vineyards in 'Hot, Very Dry' regions can be hundreds of times the size in other regions. This limitation could be overcome by incorporating the profitability of vineyards and comparing the financial success of working at different operational scales.

The second limitation was the lack of further explanatory variables to help link models to causal effects. Variables such as using renewable energy, contractors, and the occurrence of disease, fire and frost were explored initially to capture the discrepancies between similar vineyards that produced different yields and crop values. However, none of these variables was significantly correlated with the response variables and did not add to model accuracy, even when considered as interactions. Allowance for nonlinear relationships, specifically through splines, resulted in more normally distributed residuals but at a drastically reduced overall accuracy when comparing $R^2$ and Residual Square Error. Attempts to fully explain minor variations were always overshadowed by the dramatic differences in regional trends. More data for each region would also be beneficial, allowing for better comparisons between regions.

Using other models, such as random forests and decision trees, alongside more variables and data may help uncover the reasons for under or overestimation. The use of alternative sustainable practices in the field could cause these differences. Moreover, while there is evidence to suggest that environmentally sustainable practices can reduce costs and increase efficiency whilst improving the quality of grapes, more research is needed to link these benefits across different regions and climates [10,12,13].

## Conclusion

This study delved into the relationships between resource use, grape sales price and yield. The findings underscore the multifaceted nature of vineyard management, where the interplay of size, resource allocation, climate, and regional influences collectively shape both the expected sale price and the quantity of grape yields. The average sales price of grapes was not solely tied to the overall expenditure of resources but rather to the efficient allocation of resources per area. This emphasises that factors beyond sheer scale contribute significantly to the final sale price of grapes produced. Moreover, regional and yearly variations substantially affected vineyard outputs, impacting sales price and quantity. Building on these insights, future research could link these types of interpretable models to other studies through the use of machine learning techniques and causal inference to improve accuracy and link sustainable practice would greatly strive toward the creation of decision support tools to further inform winegrowers of sustainable options and their efficacy.

## Supporting information

**Table S1.** P-values for the Pearson correlation coefficients of the non-transformed water used variable with other variables. This table highlights the statistical significance of correlations in the dataset.
(PDF)

**Fig S1.** Quantile-Quantile (QQ) plot of Model 1 residuals, showing how closely the residuals align with a normal distribution. Deviations from the line indicate departures from normality.
(TIFF)

**Fig S2.** Residuals vs. fitted values plot for Model 1, illustrating the linearity and homoscedasticity of the residuals.
(TIFF)

**Fig S3.** Quantile-Quantile (QQ) plot of Model 2 residuals, showing how closely the residuals align with a normal distribution. Deviations from the line indicate departures from normality.
(TIFF)

**Fig S4.** Residuals vs. fitted values plot for Model 2, illustrating the linearity and homoscedasticity of the residuals.
(TIFF)

**Fig S5.** Quantile-Quantile (QQ) plot of Model 3 residuals, showing how closely the residuals align with a normal distribution. Deviations from the line indicate departures from normality.
(TIFF)

**Fig S6.** Residuals vs. fitted values plot for Model 3, illustrating the linearity and homoscedasticity of the residuals.
(TIFF)

**Fig S7.** Quantile-Quantile (QQ) plot of Model 4 residuals, showing how closely the residuals align with a normal distribution. Deviations from the line indicate departures from normality.
(TIFF)

**Fig S8.** Residuals vs. fitted values plot for Model 4, illustrating the linearity and homoscedasticity of the residuals.
(TIFF)

**Fig S9**. Quantile-Quantile (QQ) plot of Model 5 residuals, showing how closely the residuals align with a normal distribution. Deviations from the line indicate departures from normality.
(TIFF)

**Fig S10**. Residuals vs. fitted values plot for Model 5, illustrating the linearity and homoscedasticity of the residuals.
(TIFF)

## Author contributions

**Conceptualization:** Bryce Boyd, Mardi Longbottom, Kerrie Mengersen.

**Data curation:** Bryce Boyd.

**Formal analysis:** Bryce Boyd.

**Investigation:** Bryce Boyd.

**Methodology:** Bryce Boyd, Kerrie Mengersen.

**Software:** Bryce Boyd.

**Supervision:** Kate Helmstedt, Mardi Longbottom, Kerrie Mengersen.

**Validation:** Bryce Boyd.

**Visualization:** Bryce Boyd, Kerrie Mengersen.

**Writing – original draft:** Bryce Boyd.

**Writing – review & editing:** Kate Helmstedt, Mardi Longbottom, Kerrie Mengersen.

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
