## [Decision Letter · Decision Letter 0]

Nov 16 2024

PONE-D-24-20245The influence of resource use on yield versus sale price

trade-off in Australian vineyardsPLOS ONE

Dear Dr. Polley,

Thank you for submitting your manuscript to PLOS ONE. After careful consideration, we feel that it has merit but does not fully meet PLOS ONE’s publication criteria as it currently stands. Therefore, we invite you to submit a revised version of the manuscript that addresses the points raised during the review process.

We look forward to receiving your revised manuscript.

Kind regards,

Alejandro F. Mac Cawley, Ph.D.

Academic Editor

PLOS ONE

3. You indicated that ethical approval was not necessary for your study. We understand that the framework for ethical oversight requirements for studies of this type may differ depending on the setting and we would appreciate some further clarification regarding your research. Could you please provide further details on why your study is exempt from the need for approval and confirmation from your institutional review board or research ethics committee (e.g., in the form of a letter or email correspondence) that ethics review was not necessary for this study? Please include a copy of the correspondence as an ""Other"" file.

“The corresponding author receives a scholarship stipend as part of their PhD program. This is funded by the Food Agility CRC.”

“Mardi Longbottom is employed by Sustainable Winegrowing Australia.”

6. We noted in your submission details that a portion of your manuscript may have been presented or published elsewhere. [Some of the same vriables such as water use, vineyard size and yield have been used along with further financial data and other environmental variables as part of a future research piece to model vineyard factor's interactions and how they relate to operating cost and revenue.] Please clarify whether this publication was peer-reviewed and formally published. If this work was previously peer-reviewed and published, in the cover letter please provide the reason that this work does not constitute dual publication and should be included in the current manuscript.

Additional Editor Comments:

Dear Authors,

The reviewers are optimistic about the paper. However, they raise some issues that need to be addressed:

1. PONE has a policy requiring authors to make all data underlying the findings described in their manuscript available without restriction, with rare exceptions.

2. Reviewers require more detail on the data, results, and correlations.

3. Look into reviewer 2's proposal to divide the vineyards into clusters that share some similarities, such as size, geographical indicators, etc.

4. A discussion on the limitations of the results and further research is needed.

5. Proof check the document.

Reviewers' comments:

Reviewer's Responses to Questions

**Comments to the Author**

1. Is the manuscript technically sound, and do the data support the conclusions?

Reviewer #1: Yes

Reviewer #2: Yes

Reviewer #3: Yes

2. Has the statistical analysis been performed appropriately and rigorously? 

Reviewer #1: I Don't Know

Reviewer #2: Yes

Reviewer #3: Yes

3. Have the authors made all data underlying the findings in their manuscript fully available?

Reviewer #1: No

Reviewer #2: No

Reviewer #3: Yes

4. Is the manuscript presented in an intelligible fashion and written in standard English?

Reviewer #1: Yes

Reviewer #2: Yes

Reviewer #3: Yes

5. Review Comments to the Author

Reviewer #1: Please provide the dataset underlying your publication. Anonymized data should be provided for the factors yield, area harvested, year, GI region (anonymization: regions 1-55), water used, scope one emissions (subset 1) and average sale prices (subset 2).

Lines 63-64: please provide more information about how and in which occasion winegrowers provided their data in the web interface.

Lines 73-74: Please provide reference or provide regional average prices, which were used to complete the dataset.

Lines 123-124: Which limits were set in order to group the different regions into cool, hot, mild, warm, or damp, dry, very dry regions? Please add the information here.

Line 190: Please correct p=0.0.39

Fig. 4: Please convert into absolute numbers (yield [t ha-1]; average price [AUD ha-1]) to make results more transparent

Line 240: Please add that the strong negative correlation between vineyard size and average sale price could be due to regionally differing vineyard sizes - therefore please cancel lines 356-358.

Line 248: Please comment that scope one emissions are only responsible for a small portion of energy input in vineyards and other inputs significantly contribute to the carbon footprint of grape production. Please add further references (e.g. Rugani et al. 2013).

Please comment on the relation between temperature and yields in the discussion and add respective references for it.

Please insert a Figure caption for figure “yield verse value”. Are yields used here scaled to area? Please use absolute numbers [t ha-1]. Why did you choose to correlate yield*average price to it? Please use absolute numbers for depicting the correlation in order to make your findings more transparent.

Reviewer #2: The paper analyzes a database of 10 years and 1261 Australian vineyards. The authors seek to establish a relationship between resource use, yield, and sale price. The authors use five general linear models to capture the relationship of resources related to water and fuel use with the output yield and average sale price of the resultant product, taking into account the size and location of the vineyard. The R2 of the models are, in general, quite high. However, there are so many aspects of the problem that it is difficult to arrive at very useful conclusions, which would need to be more specific. It seems that it might be better to try to divide the vineyards into clusters that share some similarities, such as size, geographical indicators, etc. This might lead to more useful conclusions for the vineyard managers, like certain strategies that could be recommended for vineyards of a certain cluster.

The authors should also check for typos. I found "hoiwever" instead of however.

Reviewer #3: In this paper, the authors analyze the relationship between resource use, yield, and sales price using statistical models. Specifically, they unveil an interesting interplay between size, resource allocation, climate, and regional influences in shaping sales price and yield. Although the research topic is promising, and the paper meets the criteria for publication in PLOS ONE, it must be improved in some ways (just minor changes) to be publishable. Particularly:

1. In the introduction section, establish the contributions of this paper.

2. In the methods section, justify the choice of methods employed. A framework chart specifying inputs and output may be helpful to include.

3. In the result section, provide the p-values for all coefficients in Table 5 and reference properly the figures and tables provided in the Appendix.

4. In the discussion section, provide some practical implications of this study for winegrowers and wine supply chain management.

5. Finally, in the conclusion section, provide some promising subjects for further investigation.

6. PLOS authors have the option to publish the peer review history of their article (what does this mean?). If published, this will include your full peer review and any attached files.

Reviewer #1: No

Reviewer #2: No

Reviewer #3: **Yes: **Mauricio Varas

---

## [Author Response · Author response to Decision Letter 1]

25 Feb 2025

(attached as 'Response to reviewers.pdf')

Response to Reviewers

We are grateful to the reviewers and the editor for their detailed and thoughtful feedback. Each suggestion has helped strengthen the manuscript, and we have made extensive revisions in response to all comments. The updated manuscript is now more clearly connected to vineyard outcomes and climate considerations, benefiting significantly from this review process. We believe these improvements enhance the manuscript’s overall contribution to understanding the trade-offs in resource use and economic productivity within Australian winegrowing regions.

Below, we address each comment individually, outlining the actions taken in response and specifying changes made to the manuscript. All page and line references relate to the revised manuscript (marked-up copy).

Editor’s Comments

We appreciate the editor's guidance and attention to formatting details, which have helped ensure our manuscript aligns with PLOS ONE's style requirements.

Please ensure that your manuscript meets PLOS ONE's style requirements, including those for file naming. The PLOS ONE style templates can be found at:

Formatting Sample - Main Body

Formatting Sample - Title, Authors, Affiliations

Response:

In compliance with PLOS ONE's style requirements, we have made the following formatting changes:

Updated all figure references from "Figure" to "Fig" to align with journal standards.

Arranged tables and graphs to appear immediately after the paragraphs in which they are first mentioned, ensuring smoother readability and compliance with journal style.

Adjusted the capitalisation in headings to meet the formatting guidelines.

Added the “S” prefix to supplementary materials in the appendix as suggested by Reviewer 3 and as per PLOS ONE’s requirements.

Thank you for bringing these to our attention.

Please provide additional details regarding participant consent. In the ethics statement in the Methods and online submission information, please ensure that you have specified (1) whether consent was informed and (2) what type you obtained (for instance, written or verbal, and if verbal, how it was documented and witnessed). If your study included minors, state whether you obtained consent from parents or guardians. If the need for consent was waived by the ethics committee, please include this information.

Response:

Data was not collected as part of this study and was only analysed. Participant consent is managed by the Australian Wine Research Institute (AWRI), which serves as the data custodian. Consent is informed, and participants agree to the terms and conditions provided by Sustainable Winegrowing Australia when signing up as part of the program, which includes data privacy and usage policies. The specific terms and conditions regarding data handling and privacy can be reviewed at the following links:

Sustainable Winegrowing Australia Terms and Conditions

Sustainable Winegrowing Australia Privacy Policy

These documents outline the informed consent process and data privacy practices in accordance with AWRI’s guidelines. No minors, animals, persons or biohazards were involved in the analysis.

You indicated that ethical approval was not necessary for your study. We understand that the framework for ethical oversight requirements for studies of this type may differ depending on the setting and we would appreciate some further clarification regarding your research. Could you please provide further details on why your study is exempt from the need for approval and confirmation from your institutional review board or research ethics committee (e.g., in the form of a letter or email correspondence) that ethics review was not necessary for this study? Please include a copy of the correspondence as an ""Other"" file.

Response:

Thank you for the concern regarding ethics. The study involved no direct interaction with human participants, animals or biohazards and does not present any sensitive or identifiable personal data.

We have reached out to our ethics committee and included further correspondence from them regarding this study’s exemption (ethics.eml and ethics.pdf).

Thank you for stating the following financial disclosure: “The corresponding author receives a scholarship stipend as part of their PhD program. This is funded by the Food Agility CRC.” Please state what role the funders took in the study. If the funders had no role, please state: "The funders had no role in study design, data collection and analysis, decision to publish, or preparation of the manuscript." If this statement is not correct you must amend it as needed. Please include this amended Role of Funder statement in your cover letter; we will change the online submission form on your behalf.

Response:

We have included this amended statement regarding the role of the funders in the cover letter, as requested.

Thank you for stating the following in the Competing Interests section:

“Mardi Longbottom is employed by Sustainable Winegrowing Australia.”

Response:

Although not used directly in this study, the data supporting this study contains sensitive financial and identifying information and, therefore, is not publicly available. This part of the terms and conditions mentioned previously is outlined here. Unfortunately, as part of accessing it a non-disclosure agreement is required. However, researchers may request access to the dataset through Sustainable Winegrowing Australia, which manages the data. For access inquiries, please contact Sustainable Winegrowing Australia via the contact page. The corresponding author, Bryce Boyd, is also available to assist in facilitating data access for researchers interested in further analyses. Mardi Longbottom has also extended their support in assisting other researchers access this data with high hopes that further research collaborations can be conducted. These email addresses are persistent to connect with the SWA should they cease being associated with the organisation.

We noted in your submission details that a portion of your manuscript may have been presented or published elsewhere. [Some of the same variables such as water use, vineyard size and yield have been used along with further financial data and other environmental variables as part of a future research piece to model vineyard factor's interactions and how they relate to operating cost and revenue.] Please clarify whether this publication was peer-reviewed and formally published. If this work was previously peer-reviewed and published, in the cover letter please provide the reason that this work does not constitute dual publication and should be included in the current manuscript.

Response:

No part of this current manuscript has been published or submitted for publication elsewhere. No analyses, text, or research questions overlap with any other work. A different manuscript using some overlapping variables from the same dataset (such as water use, vineyard size, and yield), has been submitted to another journal and is currently undergoing peer review. The focus of that study is on statistical relationships between vineyard factors and their direct impact on operational costs and revenue, utilising machine learning to model these interactions.

The current manuscript, however, emphasises the environmental and productivity trade-offs in Australian winegrowing regions, integrating considerations of resource use, climate impacts, and sustainability outcomes. Therefore, this manuscript does not constitute dual publication but rather complements the previous research.

We trust this clarifies the distinct focus and contribution of the current manuscript. We are happy to provide a confidential draft of the other manuscript if the editor requires this.

Please include captions for your Supporting Information files at the end of your manuscript, and update any in-text citations to match accordingly. Please see our Supporting Information guidelines for more information: http://journals.plos.org/plosone/s/supporting-information.

Response:

We updated the captions in the appendix to conform with PLOS ONE’s requirements. Each caption now includes the preceding “S” and is renumbered starting from 1. All in-text citations for these files have been updated accordingly.

Reviewers’ Comments:

Reviewer 1

Please provide the dataset underlying your publication. Anonymized data should be provided for the factors yield, area harvested, year, GI region (anonymization: regions 1-55), water used, scope one emissions (subset 1) and average sale prices (subset 2).

As stated in the cover letter and above, the data cannot be shared publicly as it is third-party data that contains sensitive finance and identity information. The authors of this manuscript do not own or have any rights to the data and have signed non-disclosure agreements to handle the data. In line with the third-party data guidelines of PLOS One the data is available from Sustainable Winegrowing Australia, who can be reached through their website contact details here. The corresponding authors Bryce Polley and Mardi Longbottom are willing to help in further connecting researchers to this data to help improve food security and technological advancements that may be gained from further analyses. To support this, Dr Longbottom has also added their email to the contact list.

The authors would also like to note that prior researchers have been successful in accessing this dataset when liaising with Sustainable Winegrowing Australia. And further partnerships have begun to develop between other researchers and institutions beyond QUT to do further analyses on the data that Sustainable Winegrowing Australia have collected.

Lines 63-64: please provide more information about how and in which occasion winegrowers provided their data in the web interface.

We have expanded the Methods section to include further details on how and when winegrowers entered their data into the web interface. Data was recorded manually by winegrowers via a web-based interface provided by Sustainable Winegrowing Australia (SWA). We also referenced the SWA user manual, which outlines the data entry process and interface.

New text (Line 78): Data recorded by Sustainable Winegrowing Australia were entered manually by winegrowers voluntarily using a web-based interface. Data was recorded after the Australian crush (February to April), with entries due for the season by the 31st of August each year. The SWA user manual outlines details surrounding the web interface and its requirements [32]. Vineyards volunteered data as part of Sustainable Winegrowing Australia initiatives, which included collaboration, workshops and certification of sustainable practices using third-party auditors. Vineyards were only included for each model if they recorded all the variables used for the corresponding model (see Table 1). Each vineyard had at least recorded region, harvest year, yield and area harvested. Other variables used but not present for every vineyard were average sale price, water used, and fuel used (diesel, petrol, biodiesel and LPG). Fuel use was converted to equivalent tonnes of carbon dioxide and collectively referenced as emissions to enable direct comparisons between fuels. All variables were continuous except for harvest year and region, which were categorical variables (Table 1).

Lines 73-74: Please provide reference or provide regional average prices, which were used to complete the dataset.

We have clarified in the manuscript that regional average prices used to complete the dataset were sourced from Wine Australia’s publicly available data. We have included references to the relevant Wine Australia Annual Reports to support this data source.

New text (Line 94): As data from Sustainable Wine Australia were voluntarily given, missing values were improved using regional average prices from the Wine Australia (previously the Winemakers Federation of Australia) data. Data from Wine Australia were collected via phone surveys and included total tonnes purchased, the average cost per tonne, and yearly change in price for region and grape varietal; the data is publicly available through the Wine Australia Annual reports [19, 20, 33–41].

Lines 123-124: Which limits were set in order to group the different regions into cool, hot, mild, warm, or damp, dry, very dry regions? Please add the information here.

We have added a description of the classification criteria and described the process undertaken for their derivation. We further add the reference to the work where this was done Coombe & Dry 2010. We also further guide readers to the Wine Australia climate atlas that documents many supplement measures for every GI Region.

New text (Line 149): Each GI Region's climatic properties were summarised using predefined classifications per the [32] user manual. The user manual describes climates by rainfall and temperature, creating supersets of regions of similar climatic properties. The climatic groups illustrated similarities and differences in sets larger than GI Regions.

These classifications were literal descriptors used by industry members to discern between different weather types within winegrowing regions. However, the climate descriptors for Australian GI regions were originally conceived by [49] using temperature-based indices. A key metric employed was the Mean January Temperature (MJT), which indicates heat accumulation during the growing season. Regions were categorised based on their MJT values into cool, intermediate, warm, and hot climates. This classification was used to aid viticulturists in selecting grape varieties best suited to the thermal conditions of each region. Further in-depth summarisations of Australian wine region climates can be found in greater detail using Wine Australia's climate atlas [50]. Other climatic descriptors were explored, such as the Köppen climate classification; however, this index did not offer enough granularity to highlight any trends or clustering within the data."

Line 190: Please correct p=0.0.39.

Thank you for pointing this out. We have corrected the p-value as requested.

New text (Line 254): The three exceptions were scope one emissions in Model 3 (p=0.22) and Model 4 (p=0.39) and the interaction between the area harvested and water used in Model 2 (p=0.22)

Fig. 4: Please convert into absolute numbers (yield [t ha⁻¹]; average price [AUD ha⁻¹]) to make results more transparent.

We have updated the axes on Figure 4 to display absolute values to improve clarity and transparency in presenting the results.

Line 240: Please add that the strong negative correlation between vineyard size and average sale price could be due to regionally differing vineyard sizes – therefo

---

## [Decision Letter · Decision Letter 1]

Nov 16 2024

The influence of resource use on yield versus sale price

trade-off in Australian vineyards

PONE-D-24-20245R1

Dear Dr. Polley,

We’re pleased to inform you that your manuscript has been judged scientifically suitable for publication and will be formally accepted for publication once it meets all outstanding technical requirements.

Within one week, you’ll receive an e-mail detailing the required amendments. When these have been addressed, you’ll receive a formal acceptance letter and our manuscript will be scheduled for publication.

Kind regards,

Alejandro F. Mac Cawley, Ph.D.

Academic Editor

PLOS ONE

Additional Editor Comments (optional):

I am pleased to confirm that your paper "The influence of resource use on yield versus sale price trade-off in Australian vineyards" has been accepted for publication in Plos One.

Reviewers' comments:

Reviewer's Responses to Questions

**Comments to the Author**

1. If the authors have adequately addressed your comments raised in a previous round of review and you feel that this manuscript is now acceptable for publication, you may indicate that here to bypass the “Comments to the Author” section, enter your conflict of interest statement in the “Confidential to Editor” section, and submit your "Accept" recommendation.

Reviewer #2: All comments have been addressed

Reviewer #3: All comments have been addressed

2. Is the manuscript technically sound, and do the data support the conclusions?

Reviewer #2: Yes

Reviewer #3: Yes

3. Has the statistical analysis been performed appropriately and rigorously? 

Reviewer #2: Yes

Reviewer #3: Yes

4. Have the authors made all data underlying the findings in their manuscript fully available?

Reviewer #2: No

Reviewer #3: No

5. Is the manuscript presented in an intelligible fashion and written in standard English?

Reviewer #2: Yes

Reviewer #3: Yes

6. Review Comments to the Author

Reviewer #2: The authors have done a very good job answering the questions and comments I noted in the first review. I have no further observations to the paper.

Reviewer #3: All of my concerns have been adequately addressed, and I therefore recommend acceptance of the paper.

7. PLOS authors have the option to publish the peer review history of their article (what does this mean?). If published, this will include your full peer review and any attached files.

Reviewer #2: **Yes: **Sergio Maturana

Reviewer #3: No

---

## [Editor Report · Acceptance letter]

PONE-D-24-20245R1

PLOS ONE

Dear Dr. Boyd,

I'm pleased to inform you that your manuscript has been deemed suitable for publication in PLOS ONE. Congratulations! Your manuscript is now being handed over to our production team.

Kind regards,

on behalf of

Dr. Alejandro F. Mac Cawley

Academic Editor

PLOS ONE